# The Effectiveness of a 10-Methacryloyloxydecyl Dihydrogen Phosphate (10-MDP)-Containing Hydrophilic Primer on Orthodontic Molar Tubes Bonded under Moisture Contamination: A Randomized Controlled Trial

**Ahmed Abduljawad [1],*, Harraa Mohammed-Salih [1],*, Majid Jabir [2] and Ahmed Almahdy [3]**

[1] Orthodontic Department, College of Dentistry, University of Baghdad, Baghdad 10071, Iraq
[2] Applied Science Department, University of Technology-Iraq, Baghdad 10066, Iraq; 100131@uotechnology.edu.iq
[3] Department of Pediatric Dentistry and Orthodontics, College of Dentistry, King Saud University, Riyadh 11545, Saudi Arabia; almahdy@ksu.edu.sa
* Correspondence: ahmed.abdallah1203a@codental.uobaghdad.edu.iq (A.A.); dr.harraa_sabah@codental.uobaghdad.edu.iq (H.M.-S.)

**Abstract:** The debonding of orthodontic attachments adversely affects orthodontic treatment. This randomized controlled trial aims to compare the effectiveness of 10-MDP-containing hydrophilic primers under moisture contamination and hydrophobic primers under non-contaminated conditions. One hundred thirty-six molar tubes of thirty-four recruits were randomly bonded in a split-mouth cross-quadrant design. For the control group, a conventional hydrophobic primer on non-contaminated enamel was used; tubes of the test group were bonded on saliva-contaminated enamel using a 10-MDP-containing hydrophilic primer. The number of debonded molar tubes and their survival rates were recorded after a six-month follow-up. A chi-square test compared the number of failures and adhesive remnant index scores, using the Kaplan–Meier test for survival rates and Multinomial logistic regression to detect the influence of covariates. Thirty-two patients completed the trial; out of 128 tubes, 10 debonded within six months, the conventional primer scored eight failures with an 87.5% survival rate, and the 10-MDP-containing hydrophilic primer scored two failures with a 96.9% survival rate. The difference in survival rates and the adhesive remnant index between the two groups were statistically significant. Age and gender did not show a statistically significant influence on the number of bond failures. 10-MDP-containing hydrophilic primers may reduce bond failures and increase survival rates, especially in poorly isolated conditions.

**Keywords:** 10-MDP; hydrophilic primers; Assure Plus; orthodontic bond; bond failure; molar tube; moisture contamination

## 1. Introduction

There are several drawbacks associated with using orthodontic bands, including the orthodontist's increased workload and the patient's increased discomfort. Decalcification occasionally occurred under the bands, and interproximal gaps needed to be closed at the end of treatment. Thus, the dentist should bond the attachments directly to the tooth's enamel and eliminate the need for bands [1].

Buonocore's innovative acid-etching method in 1955 and orthodontic bracket bonding by Newman in 1965 improved the overall results of orthodontic treatment [2]. Resin-based materials are used for bonding orthodontic brackets to etched enamel through a technique-sensitive process. Controlling the amount of moisture is the most critical part of this process. A dry field is essential for the success of bracket bonding. Contamination can happen after the tooth surface's etching or after the primer's application, which reduces the efficacy of the bonding process [3].

There is a chance that saliva may contaminate the etched enamel surface during the bonding process. Saliva-contaminated enamel is thought to be the most frequent cause of bond failure [4]. Saliva penetrates acid-etched enamel, reducing the surface energy and making the surface unfavorable to bonding [5]. Most porosities plug up once the etched enamel has become wet, compromising the resin's ability to penetrate and resulting in resin tags with inadequate lengths and numbers [6].

The orthodontist frequently faces the challenge of bonding orthodontic attachments in a setting with a high risk of saliva contamination [7]. Traditionally, bisphenol A-glycidyl methacrylate (Bis-GMA) adhesives need dry-etched enamel for mechanical adhesion due to their hydrophobic nature and lack of chemical adhesion [8].

Bonding orthodontic attachments to molars has a higher failure rate compared to bonding to anterior teeth [9]. The rebonding of failed orthodontic attachments significantly extends the duration of orthodontic treatment [10]. Most orthodontists aim to shorten the duration of orthodontic treatment [11]. Developing hydrophilic primers provided a possible solution to this problem [12]. The incorporation of the functional monomer 10-Methacryloyloxydecyl dihydrogen Phosphate (10-MDP) improves hydrophilic resin diffusion and adhesion via its ability to chemically bind to calcium ions or amino groups in tooth structure [13]. Numerous in vitro studies back up the claim made by the manufacturers that hydrophilic primers tolerate moisture contamination and provide a strong bond for orthodontic attachments bonded in moist environments [2,14–16].

However, because of their inherent limitations and the inability to account for several significant intraoral factors, these in vitro measurements should be interpreted cautiously, even though they offer valuable information about the bonding effectiveness of various materials [17]. To our knowledge, no previous randomized controlled trial (RCT) has evaluated the efficacy of a newly available 10-MDP-containing hydrophilic primer in reducing the bond failure rate in contaminated conditions; thus, this RCT is conducted to be more convenient for clinical practice.

*Specific Objectives or Hypotheses*

The objectives of this trial were to evaluate the clinical bond failure and survival rate of molar tubes bonded using a 10-MDP-containing hydrophilic primer under contaminated conditions in comparison with a hydrophobic (conventional) primer under non-contaminated conditions (as a control) and to evaluate the effects of different arches and gender on bond failure. The null hypothesis is that there was no significant difference in the number of bond failures and survival rate of molar tubes between the two groups.

## 2. Materials and Methods

### 2.1. Trial Design and Ethical Considerations

This trial was a two-arm, split-mouth, cross-quadrant, single-center, single-operator RCT with a 1:1 allocation ratio. No changes were made to the protocol after the trial commencement. Ethical approval to conduct this study was granted before the commencement of the trial by the research ethics committee in Baghdad University/College of Dentistry, issued 594 on 4 October 2022. This trial was registered at ClinicalTrials.gov (ID): NCT05345379.

### 2.2. Participants, Eligibility Criteria, and Setting

Thirty-four consecutive patients of both genders with complete permanent dentition who needed fixed orthodontic appliances were recruited from April to September 2022. They were considered eligible for the study if they fulfilled the following inclusion criteria:

1.  Patients who required a 0.22″ metal fixed orthodontic appliance;
2.  Complete permanent dentition on both arches, with fully erupted molar teeth;
3.  Complete set of first molars with their buccal surfaces free from decay, restorations, or gingival hyperplasia;

4. No occlusal interferences that may transmit forces to the molar tube other than orthodontic forces.

The exclusion criteria were as follows:

1. Patients who have oral habits (i.e., bruxism or clenching);
2. Patients with systemic disease affecting salivary flow rate or with xerostomia;
3. Patients who have scissor bite or posterior crossbite;
4. Patients who need a molar band rather than a tube in appliance design (i.e., for expander or transpalatal arch).

All participants in the study provided their informed consent; for a minor, a parent or legal representative gave the permission, and the minor gave their assent. Without affecting the agreed-upon treatment, patients could leave the study whenever they wanted.

The study was conducted in the orthodontic department at the College of Dentistry/University of Baghdad, Baghdad, Iraq.

*2.3. Interventions*

A single-operator bonded all the molar tubes to avoid inter-operator variation. All teeth were isolated and cleaned with water and oil-free, non-fluoridated pumice using a rubber-polishing cup on a low-speed handpiece. The molars were etched for 30 s using 37% phosphoric acid: N-Etch (Ivoclar Vivadent, Schaan, Liechtenstein) (Figure 1a). With the aid of a triple syringe, the teeth were rinsed and gently dried with air until a frosty-white appearance was obtained. Isolation was performed using check retractors, saliva ejectors, and cotton rolls. Further priming steps are explained for each group as follows:

1. Control group (Hydrophobic primer, HP): an even coat of an HP primer (Transbond XT, 3M Unitek, Monrovia, CA, USA) was applied to the etched surface using a nylon bond brush. Each tooth received a gentle air blow for 2 s with the air stream aimed perpendicular to the enamel surface, followed by a 10 s light cure using a light - emitting diode, Eighteeth curing pen (Changzhou City, Jiangsu, China), with the following specification of light curing unit: light intensity—1500 mW/cm$^2$, output wavelength—380–515 nm.
2. Test group (10-MDP Hydrophilic primer, 10-MDP HP): a wet cotton roll was wiped against the etched tooth surface, and then one coat of the patient's non-stimulated saliva was taken from the upper labial sulcus (Figure 1b).

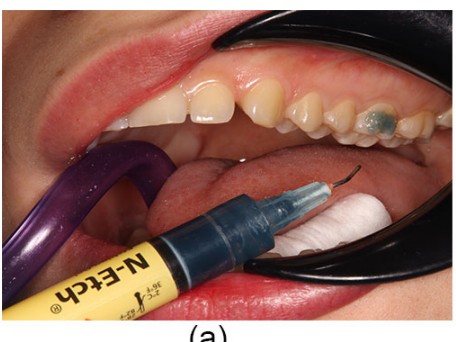
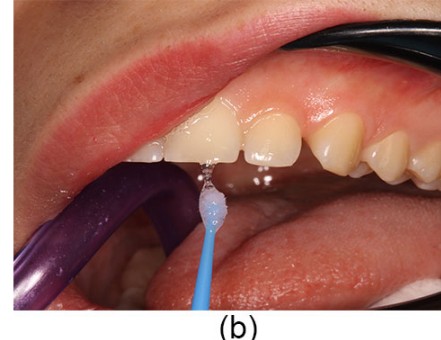

(a)                                                            (b)

**Figure 1.** Interventions: (**a**) acid etching, (**b**) the patient's non-stimulated saliva was taken from the upper labial sulcus.

The patient's non-stimulated saliva was applied to the etched and wet surface using a bond brush (Figure 2a) [14,18,19]. A liberal coat of a newly available 10-MDP HP primer (Assure Plus, Reliance, Itasca, IL, USA) was applied to the contaminated area using a nylon bond brush (Figure 2b).

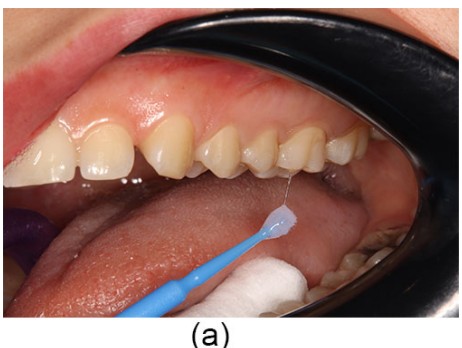
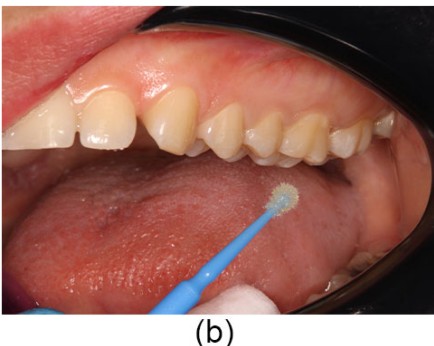

(a)                                                                  (b)

**Figure 2.** Interventions: (**a**) The patient's non-stimulated saliva was applied to the etched and wet surface, and (**b**) the 10-MDP HP primer was applied to the contaminated area.

Air was gently blown for 2 s, aiming the air stream perpendicular to the enamel surface, then light-cured for 10 s using the same device used in the control group (Figure 3a).

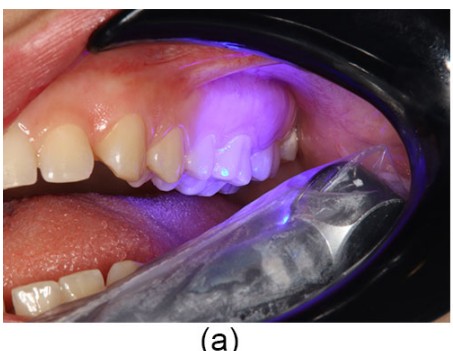
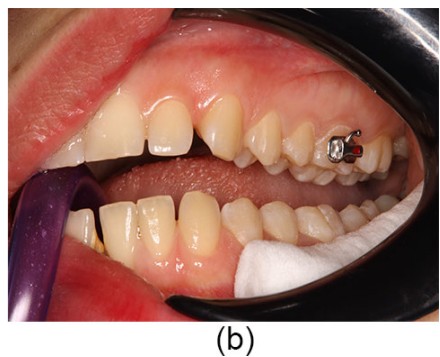

(a)                                                                  (b)

**Figure 3.** Interventions: (**a**) the enamel surface was light-cured for 10 s, (**b**) the was is positioned correctly on the buccal surface of the tooth and pressed firmly into place to express the adhesive.

Further bonding steps are the same for the two groups in the trial. After etching and priming, classic tubes 0.022" (IOS International Orthodontics Services, Capricorn St, Stafford, TX, USA) are bonded to the teeth using CONTEC lc adhesive (Dentaurum GmbH & Co. KG, Turnstr, Ispringen, Germany). The adhesive is applied to the tube's base, and the tube is positioned correctly on the buccal surface of the tooth and pressed firmly into place to express adhesive from the rim of the tube's base (Figure 3b).

The excess adhesive was removed with an explorer before curing. Then, the tube was light-cured for 20 s, 5 s each, from each tube's mesial, distal, occlusal, and gingival aspects, using the same light-curing device used in primer curing. However, before trial commencement, any occlusal interference expressed was excluded, but the molar tubes' post-bonding was further examined for any occlusal interferences; if it was found, it was discarded from the trial. In the context of the straight wire technique, initial wires are fitted 10–15 min after the completion of bonding.

Patients were instructed to perform good oral hygiene and avoid a hard diet. They were seen every four weeks for molar tube checking and appliance activation. In the event of a failure, patients were instructed to contact the operator and come in as soon as possible. The time, tooth number, and reasons for debonding were recorded; in addition, the tooth's surface was photographed under magnification using a Canon EOS 850D Professional digital camera (Tokyo, Japan) with Sigma 105 mm F2.8 EX DG OS HSM Macro Lens (Kanagawa, Japan). A magnification of 10× was used to detect the Adhesive Ruminant Index (ARI) scores; the degree of magnification was calculated depending on the following formula: magnification = size after magnification/actual size.

The failure mode was categorized depending on how much adhesive remains on the tooth surface using the index of Årtun and Bergland [20]. The principal investiga-

tor made the ARI scoring. Following each patient's recruitment, the total follow-up duration was six months.

### 2.4. Outcomes (Primary and Secondary) and Any Changes after Trial Commencement

The primary outcome was the number of bond failures over six months to evaluate the clinical bond failure and survival rate of molar tubes using a 10-MDP-containing hydrophilic primer under moisture-contaminated conditions compared with a hydrophobic primer under non-contaminated conditions.

The secondary outcomes were to evaluate the adhesive left on the tooth (ARI) and to investigate the influence of different arches and gender differences on the primary outcome. These outcomes were assessed after a 6-month follow-up period; there were no outcome changes after trial commencement.

### 2.5. Sample Size and Power of the Study

The sample size calculation is based on the primary outcome. Since the primary outcome is categorical (nominal type), the chi-square test of goodness of fit is used in the calculation, considering 80% study power, with a significance level of 0.05, to detect an effect size of 0.5 based on previous studies [21,22].

Using the G*Power 3.1.9.6 program (Franz Faul, Unikiel, Germany), the minimum sample size was 32 recruits. Considering 5% dropout raises the sample size to 34.

Interim analyses and stopping guidelines are not applicable since Patients with recurrent bond failures caused by trauma or habit would be excluded from further analysis.

### 2.6. Randomization

Using an online randomization tool [23], an allocation scheme was generated, in which one of the primers was randomly assigned to the maxillary right quadrant, and by using a cross-quadrant method, the other primer was immediately assigned. Before the trial began, the side allocation of the primers was concealed in sequentially numbered, opaque, sealed envelopes. For the sake of randomization, an independent individual opened each subsequent envelope.

### 2.7. Blinding

The subjects had no idea which primer was on which side of their mouth. As the contamination condition and the consistency of the primers were different, it was impossible to blind the operator to the kind of primer used on each quadrant of the mouth.

### 2.8. Statistical Analysis

The data were subjected to computerized statistical analysis using SPSS (Statistical Packages for Social Sciences) version 26 (IBM, Chicago, IL, USA). A *p*-value of 0.05 or less was considered statistically significant. A reliability test using an interclass correlation coefficient (ICC) was used to test intra- (after the one-month interval) and inter-examiner (with co-investigator) calibrations for 5 debonded molar tubes for ARI scores. A Pearson chi-square test was used to compare the number of bond failures. Kaplan–Meier analysis with log-rank test was used to estimate the tube survival rate. Multinomial logistic regression was used to investigate the association of covariates (arch and patient's gender) with the primary outcome, and further analysis to investigate the effects of covariates using the Cox regression model with response time to failure. Fisher's exact test was used to compare ARI scores between groups.

## 3. Results

### 3.1. Participant Flow

Forty patients were assessed for eligibility; five did not meet the inclusion criteria, and one declined to participate. For each group, thirty-four participants were randomly assigned; all received the intended treatment, and two recruits removed their orthodontic

appliances early and withdrew from the trial. Thirty-two were analyzed for the primary outcome. Figure 4 shows the CONSORT flow diagram.

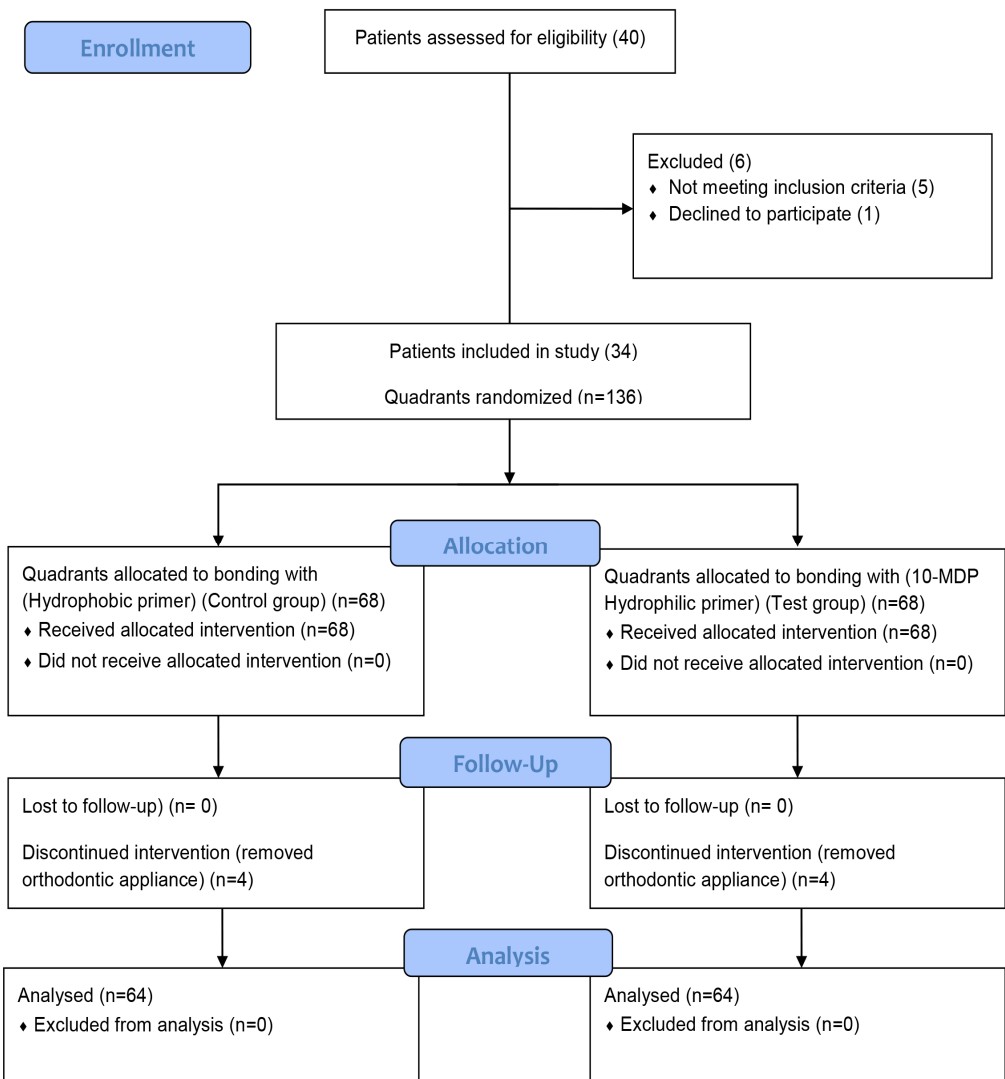

**Figure 4.** A CONSORT diagram showing patients' flow through each trial stage.

### 3.2. Recruitment

Recruitment occurred between 11 April and 1 September 2022. The follow-up duration was six months, starting from the recruitment date of every participant.

### 3.3. Baseline Data

The baseline demographic data of the sample are summarized in Table 1.

**Table 1.** The baseline demographic data of the sample.

| Participants | | 32 |
|---|---|---|
| **gender** | Male | 12 |
| | Female | 20 |
| **age** | Mean age (years) | 19 |
| | Minimum age (years) | 14 |
| | Maximum age (years) | 24 |

*3.4. Numbers Analysed*

Thirty-two of the assigned recruits carried out the analysis. For each group, sixty-four molar tubes were included in each analysis.

*3.5. Outcomes*

3.5.1. Primary Outcomes

The difference in number of bond failures between groups compared using Pearson Chi-square is significant ($p$ = 0.048). The difference in survival rates between groups compared using Kaplan–Meier analysis with log-rank test is significant ($p$ = 0.047).

The total number of debonded molar tubes within six months was 10; 40% debonded within the first month, and 80% debonded within the first three months. The frequencies and percentages of surviving and debonded molar tubes within each group are summarized in Table 2. Survival time is illustrated by the survival functions graph (Figure 5).

**Table 2.** The frequencies and percentages of each group's surviving and debonded molar tubes.

| Groups | Survival | |
|---|---|---|
| | **Survived** | **Debonded** |
| **control group (HP)** | 56 | 8 |
| | 87.50% | 12.50% |
| **test group (10-MDP HP)** | 62 | 2 |
| | 96.90% | 3.12% |
| **total** | 118 | 10 |
| | 92.18% | 7.81% |

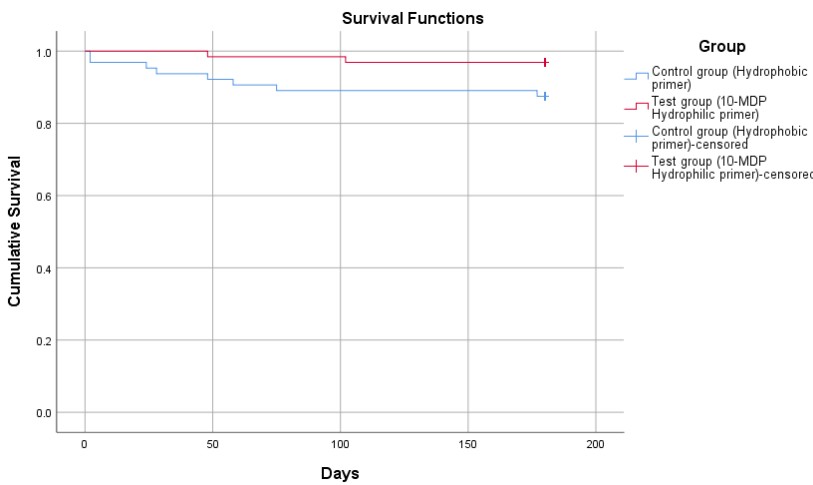

**Figure 5.** The survival functions graph illustrates molar tubes' survival in the two groups through time.

3.5.2. Secondary Outcomes

The multinomial logistic regression test and Cox regression model with response time to failure showed no statistically significant association between covariates (arch and patient's gender) and the primary outcome (Table 3).

The distribution of debonded and survived molar tubes according to gender and arch within groups is shown in Table 4.

The reliability test, ICC, showed excellent matching for intra-examiner (0.972) and inter-examiner (0.938) calibrations for ARI scores; the ARI scores of debonded tubes are summarized in Table 5.

**Table 3.** The *p*-value of the multinomial logistic regression test and Cox regression model with response time to failure.

| Covariates | Test | |
|---|---|---|
| | **Multinomial Logistic Regression (*p*-Value)** | **Cox Regression (*p*-Value)** |
| arch | 0.192 | 0.206 |
| gender | 0.603 | 0.619 |

A *p*-value equal to or less than 0.05 is considered statistically significant.

**Table 4.** The descriptive distribution of debonded and surviving molar tubes according to gender and arch within groups.

| Gender and Arch | Total Number of Bonded Molar Tubes | | Survival | | | |
|---|---|---|---|---|---|---|
| | | | Survived | | Debonded | |
| | **Control Group** | **Test Group** | **Control Group** | **Test Group** | **Control Group** | **Test Group** |
| **male** | 24 | 24 | 22 | 23 | 2 | 1 |
| | 18.80% | 18.80% | 17% | 18.00% | 1.60% | 0.80% |
| **female** | 40 | 40 | 34 | 39 | 6 | 1 |
| | 31.30% | 31.30% | 26.60% | 30.50% | 4.70% | 0.80% |
| **maxillary arch** | 32 | 32 | 29 | 32 | 3 | 0 |
| | 25.00% | 25.00% | 22.70% | 25.00% | 2.30% | 0.00% |
| **mandibular arch** | 32 | 32 | 27 | 30 | 5 | 2 |
| | 25.00% | 25.00% | 21.10% | 23.40% | 3.90% | 1.60% |

The difference in the total number of bond failures between the control and the test groups compared via the Pearson chi-square test is statistically significant (*p* = 0.048).

**Table 5.** The ARI scores of debonded molar tubes and their percentage.

| Groups | ARI Scores | | | | Total |
|---|---|---|---|---|---|
| | **0** | **1** | **2** | **3** | |
| **control group (HP)** | 7 | 1 | 0 | 0 | 8 |
| | 87.50% | 12.50% | 0.00% | 0.00% | 100.00% |
| **test group (10-MDP HP)** | 0 | 0 | 0 | 2 | 2 |
| | 0.00% | 0.00% | 0.00% | 100.00% | 100.00% |
| **total** | 7 | 1 | 0 | 2 | 10 |
| | 70% | 10% | 0.00% | 20% | 100.00% |

0, no adhesive on the tooth; 1, less than half of the adhesive on the tooth; 2, more than half of the adhesive on the tooth; 3, all adhesive left on the tooth, with an impression of the tube base. The difference in ARI scores between the test and the control groups compared using Fisher's exact test is statistically significant (*p* = 0.022).

Examples of some debonded molar tubes with their respective molar surfaces and ARI scores are shown in Figure 6. The difference in ARI scores between the test group and the control group is statistically significant (*p* = 0.022).

*3.6. Harms*

The trial caused no harm.

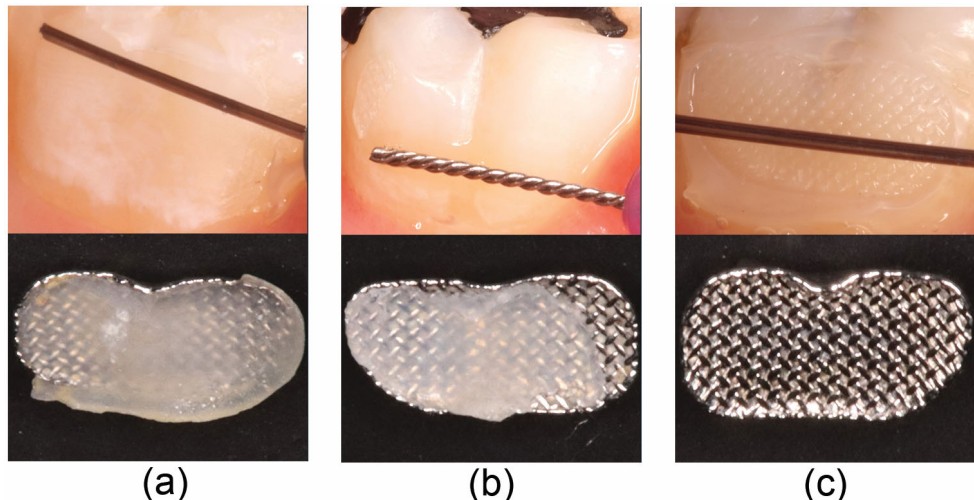

**Figure 6.** Debonded molar tubes with their respective molars and ARI scores: (**a**) no adhesive on the tooth (score 0), (**b**) less than half of the adhesive on the tooth (score 1), and (**c**) all adhesive left on the tooth, with an impression of the tube base (score 3). The difference in ARI scores between the test and the control groups compared using Fisher's exact test is statistically significant ($p = 0.022$).

## 4. Discussion

### 4.1. The Number of Bond Failures and the Tubes' Survival Rates

From a statistical point of view, the null hypothesis of this study has been rejected. The total number of debonded molar tubes within six months is 10 out of 128. The control (HP group) scored eight bond failures with an 87.5% survival rate, while the test group (10-MDP HP) scored two failures with a 96.9% survival rate. Concerning the number of debonded tubes and the tubes' survival rates, both the differences between the control and the test group were significant.

The higher number of bond failures in the control group reflects the influence of some clinical factors on the bond strength of hydrophobic primers that are absent in in vitro studies, such as intraoral relative humidity and gingival moisture crevicular fluid; these factors might have made dry enamel wet. The range of intraoral relative humidity without a rubber dam is between 78% and 94% when the temperature is between 26 °C and 29 °C; relative humidity in the mouth cannot be effectively controlled without a rubber dam [24]. Gingival crevicular fluid increases the number of bond failures [25]. Simple moisture exclusion using cotton rolls alone is unsuitable for dry-field techniques because of the 100% relative humidity [26,27]. Thus, saliva ejector and cotton roll isolation alone cannot maintain a 100% dryness of the enamel surface.

The fact that weaker shear bond strength can be obtained by using a hydrophobic primer under contamination has been noted by previous in vitro studies [2,15,28–30]. This hydrophobicity is attributed to its composition of 2, 2-bis[p-(20-hydroxy-30-methacryloxypropoxy) phenyl] propane/triethyleneglycol dimethacrylate (Bis-GMA/TEGDMA) with a ratio of 1/1 by weight. The structure of BIS-GMA has a highly hydrophobic nature [31]. The more weight of Bis-GMA and the less TEGDMA in the primer, the larger the contact angles and the greater the hydrophobic properties and vice versa [32].

On the other hand, the test group showed a higher survival rate that might be due to the hydrophilic elements like Hydroxyethyl Methacrylate (HEMA) performing the wetting agent function, which makes it possible to achieve a reduced contact angle and accelerated molecule extension [15]. Additionally, it contains alcohol, which functions as a drying agent; it seeks out moisture, evaporates it from the bonding field, and brings the resin in, ensuring a strong bond [33]. In addition to using HEMA and alcohol in its composition, 10-Methacryloyloxydecyl dihydrogen Phosphate (10-MDP) is incorporated in the primer. Because of its acidic nature (esters of phosphoric acid), 10-MDP has the potential to etch and demineralize tooth tissues, making it a promising candidate for use in adhesives requiring

self, selective, and total etching options [34]. Furthermore, the presence of 10-MDP increases the shear bond strength of orthodontic brackets to the amalgam [35], zirconia [36–38], and titanium [39]. Compared to adhesives made with other acidic ingredients, those containing 10-MDP demonstrated superior enamel bonding performance [40].

Previous studies on the effect of contamination with saliva on the bond strength of orthodontic attachments have yielded similar and, in some cases, different results. Hadrous et al. [15] stated that hydrophilic primers, unlike conventional primers, yielded clinically acceptable shear bond strength in dry and saliva-contaminated conditions. Goswami et al. [14] found that the moisture contamination did not impact the moisture-insensitive primer's ability to maintain the SBS and adhesive on the tooth surface. Cacciafesta et al. [33] concluded that contaminating the enamel surface with saliva yielded a significant drop in the bond strength of the hydrophilic primer. However, contaminated conditions did not decrease the bond strength below the clinically acceptable level. Webster et al. [41] noticed that the reapplication of hydrophilic primer after saliva contamination did not significantly increase bond strength.

Eighty percent of bond failures occurred within the first three months of the trial. A previous study found that nearly 50% of all bond failures occurred within the first two months of therapy [42]. Another study concluded that most bond failures occurred within the first six months of the trial. Three potential causes are cited. First, any problems with the bond strength of a particular bracket/adhesive combination (due to air inclusions, insufficient enamel surface etching, or poor moisture isolation) would become apparent during this phase of treatment. Second, patients are still adapting during this treatment phase, so bonding problems can arise when they eat a restricted diet. Third, heavy occlusal forces may be applied to many of the bonded attachments during the initial phase of treatment, resulting in bond failure [43]. However, most of these potential causes were excluded from the current trial.

### 4.2. The Effect of Gender and Arch on the Number of Bond Failures and the Tubes' Survival Rates

The number of debonded molar tubes in females was greater than in males; this could be attributed to the larger number of females within the sample (simple random sampling) due to females having a greater desire to have orthodontic treatment than males [44]. Furthermore, the number of debonded molar tubes in the mandibular arch is greater than in the maxillary arch; this could be due to the increased masticatory loading on mandibular tubes from chewing hard food and poorer moisture control during the bonding of the mandibular arch [45]. However, a higher failure rate in the maxillary arch was reported by another work [46]. Within this study, gender and the dental arch did not significantly influence the survival rate and the number of debonded molar tubes, which is consistent with another study [47].

### 4.3. The Adhesive Remnant Index

The difference in the adhesive remnant index was significant. The bond failures of the control group mostly occurred at the enamel–adhesive interface (all adhesive remained on the tube), while with the test group, the bond failed at the adhesive–tube interface (all adhesive remained on the enamel). Previous studies attributed the difference in the bond failure site to the attachment material [48] and the design of the retention means on the attachment base [49]. In this trial, the same attachment material and design were used for both groups, so this difference could be attributed to a stronger bond at the enamel adhesive than at the adhesive-bracket base interface in the test group.

The findings of this randomized controlled trial hold significant clinical implications for orthodontic practice. The research demonstrates that utilizing 10-MDP-containing hydrophilic primers, even in the presence of moisture contamination, can substantially reduce the incidence of debonded molar tubes and lead to significantly higher survival rates. This suggests that orthodontists can potentially enhance the durability and success of their treatments, particularly in scenarios where maintaining a completely dry field is

challenging. From a clinical point of view, the success of bonding is of major importance for the success of orthodontic treatment [50]. These hydrophilic primers offer a promising solution to mitigate the common problem of bond failures, ultimately improving patient outcomes and treatment efficiency.

In the realm of orthodontic research, this study distinguishes itself by clinically comparing the performance of 10-MDP hydrophilic primers under adverse conditions to conventional hydrophobic primers in ideal circumstances. While existing research has indeed explored various bonding techniques and primer types using in vitro laboratory settings [2,14–16], this study takes a notable step forward by conducting a randomized controlled trial that directly assesses the performance of these techniques in a clinical context. By bonding molar tubes in real-life conditions and comparing the outcomes between hydrophilic and hydrophobic primers, this research bridges the gap between laboratory findings and practical clinical applications. It provides valuable insights into how these bonding methods perform when faced with the challenges of moisture contamination, offering orthodontists more clinically relevant guidance for their daily practice.

### 4.4. Limitations

The results of this study should be interpreted with caution. Performing bonding procedures with a single operator makes the results only truly attributable to his practice. Some clinicians may consider a difference of less than 5 percent clinically significant. Blinding of the operator was impossible because of the different primers' consistencies and bonding conditions. It was impossible to analyze the enamel surface changes at a high level of accuracy inside the patient's mouth after debonding, as in the in vitro study. Furthermore, the number of debonded molar tubes is small in the hydrophilic group; the inferential statistical results of ARI should be interpreted with some caution.

### 4.5. Generalization

The results of this study should be interpreted with some caution. As previously mentioned, performing a bonding procedure with a single operator makes the results only truly attributable to his practice. The results are based on one specific type of molar tube and adhesive. Further multi-operator RCTs and in vitro studies using different types of molar tubes and adhesives are needed to investigate the efficacy of 10-MDP-containing hydrophilic primers.

### 4.6. Future Work

Future studies on bonding orthodontic attachments using hydrophilic primers may employ multi-operator (operators with varying levels of clinical experience) randomized controlled trials to account for clustering and cross-over effects. Decalcification rates, enamel damage, and the amount of time needed to remove the fixed appliance(s) and any residual adhesive are just a few examples of additional factors that could be evaluated.

### 5. Conclusions

- Molar tubes bonded intraorally using a 10-MDP-containing hydrophilic primer under contaminated conditions scored fewer bond failures and higher survival rates when compared with molar tubes bonded using a conventional hydrophobic primer under non-contaminated conditions; thus, the hydrophilic primer could be useful clinically, especially in poor isolation conditions.
- The gender and arch did not significantly influence the survival and number of debonded molar tubes.
- Molar tubes bonded using conventional hydrophobic primer failed at the enamel–adhesive interface, while tubes bonded using the 10-MDP-containing hydrophilic primer tend to failed at the tube-adhesive interface.

- The findings from this study highlight the significant advantages of using a 10-MDP-containing hydrophilic primer for bonding molar tubes under potentially challenging conditions. This innovative approach resulted in remarkably fewer bond failures and higher survival rates when compared to the traditional hydrophobic primer. These insights have important implications for orthodontic practice, emphasizing the potential benefits of incorporating 10-MDP-containing hydrophilic primers to improve the longevity and reliability of molar tube bonds.

**Author Contributions:** A.A. (Ahmed Abduljawad) and H.M.-S. conceived the initial research idea. A.A. (Ahmed Abduljawad) performed the literature review. The generation of randomization papers was performed by H.M.-S. The clinical bonding procedure and follow-up were performed by A.A. (Ahmed Abduljawad). Both authors carried out the statistical analyses and interpretation of data. Both authors contributed to the writing of the manuscript. M.J. and A.A. (Ahmed Almahdy) performed the formal analysis and reviewed the final manuscript prior to submission. All authors have read and agreed to the published version of the manuscript.

**Funding:** This research received no external funding.

**Institutional Review Board Statement:** The study was conducted in accordance with the Declaration of Helsinki and approved by the Ethics Committee of Baghdad University/College of Dentistry, issued 594 on 4 October 2022. This trial was registered at ClinicalTrials.gov (ID): NCT05345379.

**Informed Consent Statement:** Informed consent was obtained from all subjects involved in the study. Written informed consent has been obtained from the patients to publish this paper.

**Data Availability Statement:** The datasets used and analyzed during the current study are available from the corresponding author upon reasonable request.

**Acknowledgments:** The authors thank Ataa Ghazi (Lecturer at Mustansiriya University/College of Dentistry) for her support and advice. They would also like to express their gratitude towards the College of Dentistry Research Center as well as the Deanship of Scientific Research at King Saud University, Riyadh, Saudi Arabia, for their support in conducting this research. The authors would like to thank the University of Baghdad/College of Dentistry/Iraq and the University of Technology/Iraq for their support.

**Conflicts of Interest:** The authors declare no conflict of interest.

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
