# Peer review of "The Effectiveness of a 10-Methacryloyloxydecyl Dihydrogen Phosphate (10-MDP)-Containing Hydrophilic Primer on Orthodontic Molar Tubes Bonded under Moisture Contamination: A Randomized Controlled Trial"

_coatings, doi:10.3390/coatings13091635_

Round 1

Reviewer 1 Report

The article given to me for review concerns a randomized controlled trial. Study designed correctly, article written correctly.

 Entry

contains all the necessary elements, well introduces to the whole study.

Material and methods

Described exhaustively and understandably. The authors provided the criteria for inclusion and exclusion from the study, the selection of the group and the appropriate size of the study groups.

Clearly described procedure for using both preparations used.

Photo 1b and 2a are overexposed, could be better quality.

All necessary elements are included: randomization, blinding, statistical analysis.

Results

Presented clearly, complete with tables and photos.

Discussion

Interesting, it contains well-cited other reports on the discussed issues.

Conclusions

Correctly drawn from the research presented.

Author Response

Dear Reviewer 1,

I trust this letter finds you well. I would like to extend my sincere appreciation for your thoughtful and constructive review of our article titled "The Effectiveness of a 10-MDP Containing Hydrophilic Primer on Orthodontic Molar Tubes Bond under Moisture Contamination: A Randomized Controlled Trial" submitted to [Coatings]. Your insightful feedback and positive comments have been both encouraging and invaluable to our research team.

I am delighted to hear that you found our study to be of value and relevance to the field of orthodontics. Your feedback has affirmed our efforts and the direction of our research. Your specific remarks on the strengths of the study are particularly motivating for us.

I also wanted to address your feedback concerning the quality of Photo 1b and 2a in our manuscript. Your keen observation is duly noted, and we are committed to enhancing the visual representation of our findings. To this end, we will be replacing Photo 1b and 2a with higher-quality images that better serve to illustrate the key aspects of our research.

Once the upgraded images are prepared, we will promptly submit the revised manuscript to [Coatings] for your consideration. We sincerely hope that you will find the improvements in line with your expectations and that you will continue to support our work through the evaluation process.

Warm regards,

Reviewer 2 Report

Dear,

Manuscript is acceptable and ok.

Author Response

Dear Reviewer 2,

I hope this letter finds you in good health and high spirits. I wanted to take a moment to express my heartfelt gratitude for your thorough review of our manuscript, "The Effectiveness of a 10-MDP Containing Hydrophilic Primer on Orthodontic Molar Tubes Bond under Moisture Contamination: A Randomized Controlled Trial," submitted to Coatings. Your positive feedback has been a source of encouragement and motivation for our research team.

Warm regards,

Reviewer 3 Report

This is an interesting study on the effectiveness of 10-MDP with hydrophilic primers. The structure of the manuscript is well organized and systematic. However, it is necessary to check the contents as follows:  

- In the introduction section, the necessity and purpose of the study are clear.

- In the research method section, the participant recruitment process and appropriate sample size are followed. In addition, the intervention process is described in detail and statistical analysis methods are appropriate.

- In the results section, please the p-value notation is inconsistent, check it. Also add to the figure.

  line 220: Please (P-value= 0.048) revise to (P = 0.048).

  line 221: Please (P-value= 0.047) revise to (P = 0.047).

  line 236: Please check the P-value in Table 3.

  Please add the P-values to Tables 4 and 5 and Figure 5.  

- In the Conclusion section, describe the final significance of this research result through the main findings. Currently, only repetitive explanations of the findings exist.

Author Response

Dear Reviewer 3,

I hope this letter finds you well. I am writing to express my sincere gratitude for your time and expertise in reviewing my article titled "The Effectiveness of a 10-MDP Containing Hydrophilic Primer on Orthodontic Molar Tubes Bond under Moisture Contamination: A Randomized Controlled Trial." Your thoughtful evaluation and valuable feedback have greatly contributed to the improvement and quality of my research.

Below I addressed your comments and the response to each one:

Comments 1: 

- In the results section, please the p-value notation is inconsistent, check it. Also add to the figure.

  line 220: Please (P-value= 0.048) revise to (P = 0.048).

  line 221: Please (P-value= 0.047) revise to (P = 0.047).

  line 236: Please check the P-value in Table 3.

Response 1: Thank you for pointing this out. I agree with this comment. Therefore, I have  

line 220:  revised to (P = 0.048).

  line 221: revised to (P = 0.047).

  line 236:  P-value in Table 3 is checked and its accurate Moreover, we would like to emphasize that the gender and arch variables are not directly compared. Instead, we have employed multinomial logistic regression and Cox regression tests to analyze these variables as covariates. These analyses generate their own distinct p-values, which are presented in table 3.

Comment 2: Please add the P-values to Tables 4 and 5 and Figure 5

Response 2: Thank you for pointing this out. we will discuss every point seperately:

Table 4: the p-value is added at the bottom of table 4 despite the fact that table 4 presents only descriptive data. The p-value resulting from comparing the number of debonded tubes between the control and test groups is indeed reported in the primary outcome section of the manuscript. This comparison is fundamental to our study's primary objective. 

Table 5: the p-value is added at the bottom of table 5 despite the fact that Table 5 in our manuscript primarily contains descriptive data only. We want to clarify that Table 5 does not present a direct comparison of the scores of the adhesive remnant index between the control and test groups. Instead, it serves the purpose of providing a summary of the ARI scores.

The p-value resulting from the comparison of the scores of the adhesive remnant index between the control and test groups is indeed reported in the secondary outcome section of the manuscript. This comparison is a crucial component of our study's secondary objectives.

Figure 5: the p-value is added at the bottom of Figure 5 despite the fact that Figure 5 in our manuscript is designed to illustrate examples of the adhesive remnants and their associated scores. It is not intended to provide inferential statistical data. Rather, its purpose is to visually depict the differences in adhesive remnants and their corresponding scores between the control and test groups.The p-value resulting from the Fisher's exact test comparing adhesive remnant index (ARI) scores between the control and test groups is indeed reported in the secondary outcomes section of the manuscript.

Comment 3: - In the Conclusion section, describe the final significance of this research result through the main findings. Currently, only repetitive explanations of the findings exist.

Response 3: Thank you for pointing this out. I agree with this comment. Therefore, I have  written a describtion of the final significance of this research result through the main findings at the end of the conclusion section.

Could you please let us know if the explanations we've made suffice or if you believe further revisions are necessary to address your concerns? Your guidance is highly valued, and we want to ensure that the manuscript meets the standards of clarity and accuracy.

Your insights and feedback are crucial in helping us refine our work, and we greatly appreciate your time and effort in reviewing our manuscript.

Reviewer 4 Report

The article is suitable for publication in Coating.The topic is interesting and currently debated.

The main question of this research is to evaluate the efficacy of newly available 10-MDP containing hydrophilic primer in reducing bond failure rate in contaminated conditions The topic is interesting and original in the field. The research methodology was applied correctly and carefully. The conclusions consistent with the evidence and arguments presented. All the reference are appropriates.

Authors should include a paragraph explaining the clinical applications of the research, and an other paragraph explaining the subject area compared with other published research.

finally, high resolution images should be inserted

I belive that with small modify the article could be accepted for publishing

Author Response

Dear Reviewer 5,

I hope this letter finds you well. I am writing to express my sincere gratitude for your time and expertise in reviewing my article titled "The Effectiveness of a 10-MDP Containing Hydrophilic Primer on Orthodontic Molar Tubes Bond under Moisture Contamination: A Randomized Controlled Trial." Your thoughtful evaluation and valuable feedback have greatly contributed to the improvement and quality of my research.

Below are your comments and the responses to every one:

Comment 1: Authors should include a paragraph explaining the clinical applications of the research, and an other paragraph explaining the subject area compared with other published research.

Response 1: Thank you for pointing this out. I agree with this comment. Therefore, I have included these paragraphs at the end of discussion section.

Comment 2: high resolution images should be inserted

Response 2: Thank you for pointing this out. I agree with this comment. Therefore the images will be replaced with higher resolution.

Once again, thank you for your time, effort, and positive feedback. Your contributions are invaluable, and I am grateful for your support in making my research more robust and impactful. If you have any further suggestions or insights in the future, please do not hesitate to share them.